# Rational Decision-Making Agent with Learning Internal Utility Judgment

**Yining Ye**[1*], **Xin Cong**[1*†], **Shizuo Tian**[1], **Yujia Qin**[1], **Chong Liu**[1],
**Yankai Lin**[2†], **Zhiyuan Liu**[1], **Maosong Sun**[1]
[1]Tsinghua University [2]Renmin University of China

```
{yeyn23,tsz21}@mails.tsinghua.edu.cn
{congxin1995,liuzy,sms}@mail.tsinghua.edu.cn
yujiaqin16@gmail.com, tylch2@163.com, yankailin@ruc.edu.cn
```

## Abstract

With remarkable advancements, large language models (LLMs) have attracted significant efforts to develop LLM-based agents capable of executing intricate multi-step decision-making tasks. Existing approaches predominantly build upon the external performance measure to guide the decision-making process but the reliance on the external performance measure as prior is problematic in real-world scenarios, where such prior may be unavailable, flawed, or even erroneous. For genuine autonomous decision-making for LLM-based agents, it is imperative to develop rationality from their posterior experiences to judge the utility of each decision independently. In this work, we propose **RaDAgent** (Rational Decision-Making Agent), which fosters the development of its rationality through an iterative framework involving *Experience Exploration* and *Utility Learning*. Within this framework, Elo-based Utility Learning is devised to assign Elo scores to individual decision steps to judge their utilities via pairwise comparisons. Consequently, these Elo scores guide the decision-making process to derive optimal outcomes. Experimental results on the Game of 24, WebShop, ToolBench and RestBench datasets demonstrate RaDAgent's superiority over baselines, achieving about $7.8\%$ improvement on average. Besides, RaDAgent can also reduce costs (ChatGPT API calls), highlighting its effectiveness and efficiency.

## 1 Introduction

The autonomous agent (Searle, 1969; Wooldridge & Jennings, 1995; Maes, 1994; Hendler, 1999), as the long-standing pursuit of artificial intelligence (AI), is expected to possess the ability to plan, make decisions, and take actions to accomplish complex tasks autonomously. As large language models (LLMs) have undergone rapid development, showcasing remarkable capabilities (OpenAI, 2022; 2023; Touvron et al., 2023a;b; Dubey et al., 2024), many efforts have been devoted to developing LLM-based agent (Richards, 2023; Nakajima, 2023; AgentGPT, 2023; Wu et al., 2023b) to accomplish intricate multi-step decision-making tasks beyond traditional natural language processing (NLP) applications (Yao et al., 2022b; Hao et al., 2023a; Yao et al., 2023; Qin et al., 2023c; Chen et al., 2023; Qian et al., 2024). Even with these strides, existing LLM-based agents require external performance measures as *prior* to guide their decision-making process (Yao et al., 2023; Hao et al., 2023a; Sel et al., 2023; Lv et al., 2024). For instance, in Game of 24, which uses four numbers and basic arithmetic operations to obtain 24, Yao et al. (2023) heuristically design a prompt to assess the possibility of each decision to reach 24 and then choose the higher one as decisions accordingly. However, this manual-designed prompt may not provide accurate possibility, causing unreliable decision-making guidance. **The reliance on the external performance measure restricts the adaptability in real-world scenarios as it may be unavailable, flawed, or even erroneous.**

---

* Indicates equal contribution.
† Corresponding author.

When individuals make decisions, they not only rely on external measures but also draw upon their practical experience as *posterior* to form a sense of internal rationality. Human rational decision-making evolves through a dynamic process of experiential learning, encompassing trial-and-error, reflection, and reinforcement Kahneman & Tversky (2013). By experimenting with different decisions and observing their outcomes, individuals learn to reinforce behaviors that yield favorable results. This learning process involves reflective analysis, during which individuals critically evaluate their decision-making processes to identify biases and rectify mistakes. Through these iterations, individuals progressively enhance their decision-making by reinforcing behaviors that produce positive outcomes, thereby refining their judgment of the utility of each decision. Finally, individuals derive internal utility judgment which serves as the basis for evaluating the effectiveness of decisions and identifying optimal solutions (Arrow, 1959; Plott, 1973; Kahneman & Tversky, 2000).

To this end, we propose **RaDAgent** (Rational Decision-Making Agent) which learns internal utility judgment ability to achieve rationality for the agent. In RaDAgent, the internal utility judgment is constructed based on an iterative framework: (1) **Experience Exploration**: Due to the complexity of real-world tasks, the solution space may be infinite, and it is challenging to find the optimal solution efficiently. Hence, working in the Monte Carlo Tree Search (Kocsis & Szepesvári, 2006; Coulom, 2006) fashion, RaDAgent would explore potential decisions with higher utilities to find better solutions as many as possible for the following utility learning. (2) **Utility Learning**: Given a series of solutions, RaDAgent should make comparisons between them to learn their utilities. Due to the challenge of LLMs in directly providing accurate numerical utilities without prior, we design Elo-based Utility Learning which employs the Elo Rating system (Elo, 1967) to estimate the utilities through posterior comparison among explored solutions. After multiple comparisons, the Elo scores would converge to an accurate value representing its actual utility in achieving the task. Using the learned utilities as guidance, the exploration process focuses on discovering decisions with higher utilities. Consequently, the exploration of these enhanced decisions aids in further refining the utilities associated with each decision. Through the iterative utility judgment learning, RaDAgent can assess the numerical utility of explored decisions and then can judge the highest utility to derive the best solution with the superior outcome.

To validate the effectiveness of our RaDAgent, we implement it based on ChatGPT (OpenAI, 2022) and conduct extensive experiments on Game of 24 (Yao et al., 2023), WebShop (Yao et al., 2022a), ToolBench (Qin et al., 2023c) and RestBench (Song et al., 2023), which contains intricate multi-step decision tasks involving diverse scenarios. Experimental results demonstrate the superiority of our approach against several baselines by achieving about $7.8\%$ improvements on average to accomplish complex tasks. Moreover, extensive analyses show that our approach not only delivers superior solutions but also achieves greater efficiency by reducing the number of ChatGPT API calls.

Our contributions are threefold: (1) We propose RaDAgent, a rational decision-making agent that can construct its internal rationality to accomplish diverse real-world tasks, not relying on external performance measures. (2) We devise Elo-based Utility Learning which can learn internal utility judgment for the agent by learning Elo scores for each decision, selecting the optimal solution with the highest utilities. (3) Extensive experiments on the Game of 24, WebShop, ToolBench, and RestBench datasets demonstrate the effectiveness and efficiency of our proposed method against representative methods. Our source code is released in `https://github.com/OpenBMB/RaD-Agent`.

## 2 RELATED WORK

**Decision-Making Methods for LLM-based Agents** Efficient and effective decision-making ability is fundamental for LLM-based agents to the attainment of specific objectives (Yao et al., 2022b; 2023; Hao et al., 2023a; Besta et al., 2023; Sel et al., 2023). Although LLMs are pre-trained on a large-scale corpus which equips them with substantial common sense and knowledge to solve several problems, due to the complexity and diversity of realistic tasks, LLM-based agents still struggle to make multi-step decisions to solve realistic tasks. Recently, as Chain-of-Thought (Wei et al., 2023) demonstrates its capability to decompose complex questions into sequential intermediate steps, several LLM-based decision-making methods have been proposed to enhance the decision-making ability of agents. ReACT (Yao et al., 2022b) develops a variant of CoT to leverage the reasoning ability of LLMs in decision-making scenarios. Reflexion (Shinn et al., 2023) further offers a remedial approach to make LLMs reflect their failure and summarize the reason in the decision process,

and then correct their mistake in the second attempt. Based on these methods, some tree-based decision-making methods are proposed to adapt the decision-making ability of LLMs into specific tasks. Tree-of-Thought (Yao et al., 2023) proposes BFS and DFS decision-making algorithms in Game of 24, Creative Writing, and Mini Crosswords tasks. RAP (Hao et al., 2023a) applies the Monte Carlo Tree search algorithm to find a good solution in Blocksworld, Math Reasoning, and Logical Reasoning tasks. DFSDT (Qin et al., 2023c), following a similar tree search algorithm, proposes an efficient version of DFS to make decisions. However, the aforementioned methods need external performance measures to guide the decision-making process, which limits their scope of application. In this paper, we propose RaDAgent which learns the internal utility judgment ability with the Elo rating system to achieve rationality for agents to provide optimal solutions.

**Tool Learning** Recent investigations have cast illumination upon the burgeoning proficiencies exhibited by LLM-based agents in the mastery of instruments and the execution of decision-making processes within intricate contextual milieus (Qin et al., 2023b; Vemprala et al., 2023; Nakano et al., 2021; Qin et al., 2023a; Shen et al., 2023; Wu et al., 2023a; Schick et al., 2023; Hao et al., 2023b; Qian et al., 2023; Song et al., 2023; Qin et al., 2023c; Guo et al., 2024). The incorporation of external tools into the operational framework of LLM-based agents confers upon them immediate access to contemporaneous factual knowledge (Yang et al., 2023), imbues them with versatile multimodal capabilities (Gupta & Kembhavi, 2023), and empowers them with specialized proficiencies tailored to vertical domains (Jin et al., 2023). However, when confronted with real-world tasks that often require the utilization of multiple tools, LLM-based agents must engage in multi-step decision-making processes to select tools and determine their sequencing. Consequently, the ability for decision-making in tool learning scenarios becomes imperative to effectively tackle practical tasks.

## 3 PRELIMINARIES

**Markov Decision Process** We formulate the decision-making process within the agent as a finite-horizon Markov Decision Process (MDP) denoted by $\mathcal{M} = \{\mathcal{S}, \mathcal{A}, \mathcal{R}, \mathcal{T}\}$ with state space $\mathcal{S}$, action space $\mathcal{A}$, reward function $\mathcal{R}$, and transition function $\mathcal{T}$. Given a human instruction $Q$, the agent acting as the policy model $\pi$ is tasked with generating a decision sequence to accomplish $Q$. The agent $\pi$ starts from the initial state $s_0$ and takes an action $a_i$ based on the current state and subsequently arrives at the next state $s_{i+1}$ decided by the transition function $\mathcal{T}$. This process terminates until the agent accomplishes the task or exceeds the limitation of the number of actions, resulting in a decision sequence or trajectory $\tau = \{s_0, a_1, s_1, \cdots, s_N\}$ where $N$ is the number of actions. A reward of 1 is assigned by the reward function $\mathcal{R}$ at the end if the agent successfully accomplishes the task, otherwise a reward of 0 is assigned. To make sequential decisions toward accomplishing $Q$ autonomously, we argue that the agent needs to identify the utility $v_i$ of each decision $a_i$ and select those decisions with a higher value that holds the promise of yielding the most promising outcomes (i.e., reward), ultimately leading to the derivation of the final decision sequence that fulfills the requirements of $Q$.

**Elo Rating System** The Elo rating system (Elo, 1967), commonly used in competitive contexts offers a numerical estimation of the skill levels of players. It represents the skill levels of players by Elo scores and assesses the Elo scores through a series of one-to-one competitions. It assumes that each player's performance follows a Gaussian distribution ($x \sim \mathcal{N}(\mu, \sigma)$) and each comparison of two players is actually comparing between two samples from their Gaussian distributions. Through multiple comparisons, we can approximate their real skill levels by estimating their Elo scores.

Given two players $x$ and $y$, their Elo scores are denoted as $v_x$ and $v_y$, respectively. The expected superiority of $x$ against $y$ is calculated as:

$$E_{x>y} = \frac{1}{1 + e^{-\frac{v_x - v_y}{r}}} \tag{1}$$

where $r$ is the Elo coefficient.

Next, we run a competition between them to find the actual winner. We denote the result as $R_{x>y}$:

$$R_{x>y} = \begin{cases} 1, \text{if } x \text{ win}, \\ 0, \text{if } y \text{ win}, \\ 0.5, \text{otherwise} \end{cases} \tag{2}$$

---

**Algorithm 1** RaDAgent

---
1: **function** RaDAgent
2: root ← initialize an empty decision node
3:    **while** within computational budget **do**
4:       # Experience Exploration
5:       node ← root
6:       **while** node is not new-explored **do**
7:          node ← sample node based on Equation 5
8:       **end while**
9:       trace ← generate new decision trace from node based on ReAct
10:
11:       # Utility Learning
12:       **while** within comparison limitation **do**
13:          candidata ← sample an existing trace randomly
14:          result ← compare new trace with candidate based on Equation 6
15:          update Elo scores based on result according to Equation 3
16:          update exploration temperature $T$ based on Equation 8
17:       **end while**
18:    **return** the best trace based on Equation 9
19: **end function**

---

We then update their Elo scores accordingly:

$$v_x = v_x + K * (R_{x>y} - E_{x>y})$$
$$v_y = v_y + K * (R_{y>x} - E_{y>x})$$

$$(3)$$

where $K > 0$ is the update step size. After multiple comparisons, the Elo score will progressively converge to their expected skill levels.

## 4 METHODOLOGY

Our RaDAgent aims to find the decision sequence with the highest utility to accomplish complex instructions autonomously. It contains two principal phases to learn the internal utility judgment:

- *Experience Exploration*: The agent takes actions sequentially to form a decision sequence toward a feasible solution.
- *Utility Learning*: The agent makes judgments among decision sequences to assess the utility (i.e., Elo scores) of existing decision steps.

These two phases work in an iterative fashion, reinforcing each other's outcomes (as shown in Algorithm 1). In the experience exploration phase, the agent explores more potential decision sequences to find better solutions, which can encourage agents to learn the actual and accurate utility of each decision step. In the utility learning phase, the agent re-calculates the Elo score of each decision step with the newly explored decision sequence to learn the utility of each decision, and the learned utilities serve as a dynamic guide, steering subsequent experience exploration toward more promising and superior solutions. By iteratively cycling through these phases, the agent progressively evolves toward an optimal decision sequence with the highest utility to address instructions.

### 4.1 EXPERIENCE EXPLORATION

In RaDAgent, each experience exploration benefits from the previous exploration history based on Elo-based Utility Learning (§ 4.2). When exploring a new decision sequence, agents will select a decision step with a higher Elo score to explore further. Specifically, in RaDAgent, each decision step is assigned an Elo score explicitly. A decision step with higher Elo scores means that it is more likely to accomplish the instruction and thus Elo scores are used to guide the decision exploration process. Given an intermediate decision step $a$, its subsequent decision steps are denoted as $\{a_1, a_2, \cdots, a_n\}$. Given their learned Elo scores $\{v_i\}_{i=1}^n$, the probability of choosing to explore can be modified as:

$$P(a_i) = \frac{\exp(\frac{v_i}{T})}{\sum_j \exp(\frac{v_j}{T})}, \ a_i \in \{a_1, a_2, \cdots, a_n\}$$

$$(4)$$

where $T$ refers to the temperature. Note that only exploring the known decisions may cause the local optimal solution. Therefore, we define a rejection decision step $\hat{a}$ with an initial Elo score $\hat{v}$ to represent that *"The agent decides to explore a new decision"*. We add this rejection decision step into the subsequent decision steps as $\{a_1, a_2, \cdots, a_n, \hat{a}\}$ when selecting:

$$P(a_i) = \frac{\exp(\frac{v_i}{T})}{\sum_j \exp(\frac{v_j}{T})}, \ a_i \in \{a_1, a_2, \cdots, a_n, \hat{a}\} \tag{5}$$

The complete experience exploration process begins from the initial state $s_0$ and chooses the subsequent decision steps iteratively based on Equation 5 in a top-down manner. When it chooses the rejection decision step $\hat{a}$, the agent will generate a new decision sequence starting from the intermediate step $a$. In the iterative experience exploration process, those potential decision steps will be explored thoroughly, until finding the optimal solution.

## 4.2 UTILITY LEARNING

As external performance measures may be unavailable, flawed, or even erroneous, the agent should resort to their internal utility judgment ability to solve diverse tasks. To this end, we design an Elo-based Utility Learning, equipping the agent with the Elo rating system to provide a numerical utility to each decision step to guide the decision-making process.

The utility learning process (i.e., the Elo score estimation process) is conducted in a bottom-up manner. It first adjusts the Elo scores of the final decision steps of each decision sequence via pairwise comparison and then updates the Elo scores of the intermediate decision steps gradually. Once a new decision sequence is generated in the experience exploration phase, the agent will self-judge the Elo scores of existing decision steps via pairwise comparison. Given the newly generated decision sequence $\tau_n$, we first assign all decision steps of $\tau_n$ with an initial Elo score. Then, we randomly select a decision sequence $\tau_i$ from existing decision sequences $\mathbb{T} = \{\tau_1, \tau_2, \cdots, \tau_{n-1}\}$ and use agents to compare $\tau_n$ with $\tau_i$ to judge which one has the superior performance. Since the LLM-based comparison is sensitive to the candidate order (Qin et al., 2023d; Chiang & Lee, 2023; Wang et al., 2023a), we conduct comparisons twice with different orders.

$$R_{t_n > t_i} = \begin{cases} 1, \text{if } \tau_n \text{ win twice,} \\ 0, \text{if } \tau_i \text{ win twice,} \\ 0.5, \text{otherwise} \end{cases} \tag{6}$$

Getting the comparison result, we update the Elo scores of the final decision steps of $\tau_n$ and $\tau_i$ based on Equation 3. Next, we calculate the Elo scores of intermediate decision steps based on their subsequent decision steps. Specifically, given an intermediate decision step $a_i$, we calculate its Elo scores as follows:

$$v_i = \sum_{a_j \in \text{Child}(a_i)} (\alpha_j * v_j), \tag{7}$$

where $\text{Child}(a_i)$ refers to the set of the subsequent decision steps of $a_i$, $\alpha_j = \frac{\exp(v_j/T)}{\sum_k \exp(v_k/T)}$ is the normalized weight and $T$ is from Equation 5. By repeating the comparison via randomly sampling decision sequences, the Elo score of each decision step will converge to its expected value.

When guiding the experience exploration process, the Elo score of a decision step with a few number of Elo updates may not represent its real value accurately. Such a decision step cannot be fully trusted for exhaustive exploration. Hence, we adjust the temperature $T$ in Equation 5 based on the number of the Elo update. Let $M_a$ be the number of the Elo update of the decision step $a$. The temperature of $a$ is annealed as follows:

$$T_a = T_0 * \frac{1}{1 + \sqrt{\ln(M_a + 1)}} \tag{8}$$

where $T_0$ is the default temperature. With the growth of the number of Elo updates, the approximated Elo score converges to its real value. At this time, we tend to explore the most possible decision.

After engaging in extensive experience exploration and utility learning, the agent learns the internal utility judgment to construct rationality that allows it to select the best-performed one as the final

solution. Specifically, given all existing decision sequences $\mathbb{T} = \{\tau_1, \tau_2, \cdots, \tau_n\}$, the one which final decision with the highest utility is selected as the final solution.

$$t = \arg\max_{\tau \in \mathbb{T}} \{V(a_N)\} \qquad (9)$$

where $a_N$ refers to the final decision step.

# 5 EXPERIMENTS

## 5.1 EXPERIMENTAL SETTINGS

**Datasets** We conduct extensive experiments on Game of 24 (Yao et al., 2023), WebShop (Yao et al., 2022a), and ToolBench (Qin et al., 2023c) datasets. Game of 24 aims to use 4 numbers and four fundamental arithmetic operations $(+ - */)$ to reach 24. WebShop focuses on simulating the process of searching, browsing, and selecting items on an online shopping platform in order to obtain desired items. ToolBench has thoughtfully constructed a diverse and intricate collection of human instructions of over 16K APIs from 49 categories. We focused on the intra-category multi-tool instruction scenario which accurately reflects the complexities involved in real-world tasks, necessitating the use of various tools and requiring multi-step decision-making processes. We use 100, 500, and 500 instances for Game of 24, WebShop, and ToolBench to evaluate the decision-making ability respectively. Details of each task (including task description, action space, etc) can refer to Appendix A.3.

**Baselines** We compare RaDAgent with the following decision-making methods: (1) **CoT** (Wei et al., 2023; Yao et al., 2022b) decomposes reasoning into explicit intermediate steps and we adapt ReAct (Yao et al., 2022b) to make sequential decisions. (2) **CoT@3** extends CoT by running the decision-making process three times independently for an instruction and finally generates a total of three decision sequences. (3) **Reflexion** (Shinn et al., 2023) builds upon CoT@3 and allows LLMs to engage in self-reflection on their previous decision sequences. (4) **ToT-BFS** (Yao et al., 2023) constructs a decision tree in a top-down manner to search for a feasible solution. (5) **ToT-DFS** (Yao et al., 2023) constructs a decision tree by going as deep as possible along each branch and exploring the most recently visited states. (6) **DFSDT** (Qin et al., 2023c) is an improved version of DFS, which allows agents to dynamically assess different decision states and choose to either proceed along a promising path or abandon an existing state and expand another one.

**Evaluation Metrics** To ensure a rigorous and accurate evaluation of the performance of our proposed decision-making approach, we adopt three evaluation metrics for each dataset respectively: (1) **Success Rate** (Yao et al., 2023) measures the proportion of valid equations generated by the agent's arithmetic operations that yield a result of 24, using the given input numbers. (2) **Reward** (Yao et al., 2022a) evaluates the similarity (a value between 0 and 1) between the attributes of the items chosen by the agent and the attributes of the items actually purchased by human. (3) **Pass Rate** (Qin et al., 2023c) assesses the ability of agents to successfully accomplish complex real-world tasks by using tools sequentially. It calculates the proportion of instructions that an agent completes.

**Implementation Details**

We use OpenAI ChatGPT `gpt-3.5-turbo-0613-16k` to implement our approach (our designed prompt can refer to Appendix A). Our approach involves conducting a decision-exploration process 20 times and finally selecting the decision sequence with the highest Elo score as the final decision. For Elo-based Utility Learning, the initial Elo score of the decision step is set as 0.0 and the Elo coefficient $r$ is set as 173.72 according to the vanilla Elo rating system (Elo, 1967). The Elo score of $\hat{d}$ in Equation 5 is set as 0.0. $K$ in Equation 3 is set as 50. To manage the computational cost of ChatGPT API calls, we set a maximum limit of 12 steps for each decision sequence for a decision-searching process. Detailed analyses of each hyperparameter are discussed in Appendix B.

## 5.2 OVERALL RESULTS

To validate the effectiveness of our proposed RaDAgent approach, we first study whether our approach can accomplish more complex tasks. The results are shown in Table 1 and we can observe that: (1) Across all datasets, RaDAgent consistently outperforms all baselines, indicating that incorporating the utility judgment internally empowers agents to accomplish a broader range of tasks effectively.

Table 1: Main experimental results on Game of 24, WebShop, and ToolBench dataset. Bold marks the best performance.

| Model | Game of 24 | WebShop | ToolBench |
|---|---|---|---|
| CoT | 6.00 | 56.23 | 16.60 |
| CoT@3 | 7.00 | 56.45 | 31.20 |
| Reflexion | 7.00 | 57.21 | 26.60 |
| ToT-BFS | 11.00 | 50.20 | 38.00 |
| ToT-DFS | 14.00 | 55.60 | 45.58 |
| DFSDT | 29.00 | 57.25 | 50.20 |
| RaDAgent | **43.00** | **59.36** | **61.92** |

Table 2: Results on the real-world RestBench dataset. Baselines are reported from Song et al. (2023). Bold marks the best performance.

| Model | TMDB | Spotify |
|---|---|---|
| Offline | 33.0 | 36.4 |
| DEPS | 43.0 | 43.8 |
| ReAct | 57.0 | 49.1 |
| Reflexion | 59.0 | 61.4 |
| RestGPT | 79.0 | 74.5 |
| RestGPT(ChatGPT) | 65.0 | 72.3 |
| RaDAgent | **84.0** | **80.7** |

(2) In Game of 24[1] and ToolBench domains, RaDAgent exhibits the capability to assign lower elo scores to decision steps that lead to failure. Consequently, these Elo scores serve as guidance for agents to avoid such decisions and achieve success. (3) For Webshop, while our method still outperforms all baselines, it only achieves only marginal gains. This is attributed to the fact that Webshop provides only one "golden answer" for each instruction, while several other items actually meet the requirements. Consequently, these alternative items receive lower rewards as they deviate from the golden answer, resulting in an underestimation of the performance.

# 6 ANALYSIS

## 6.1 GENERALIZATION TO REAL-WORLD ENVIRONMENT

To verify that our method is robust and applicable to real-world environments, we expand our evaluation to the RestBench dataset (Song et al., 2023), which features restful APIs from two prominent real-world applications: TMDB and Spotify. All APIs in RestBench will be authentically called and get the **real-time**, **dynamic**, and **unpredictable** responses from the TMDB and Spotify platform, revealing a more challenging scenario for decision-making. This dataset includes human-annotated real-world tasks with ground truth decision sequences. We compare our method with Offline (Qin et al., 2023b), DEPS (Wang et al., 2023b), ReAct (Yao et al., 2022b), Reflection (Shinn et al., 2023), RestGPT (Song et al., 2023) and its ChatGPT version. Following RestGPT (Song et al., 2023), we report the **Correct Path Rate** which calculates the proportion of the correct decision sequences. Experimental results are shown in Table 2, which have demonstrated that our approach outperforms the baselines and achieves the best Correct Path Rate (84.0% and 80.7% for TMDB and Spotify respectively). This result underscores the effectiveness of our method in making decisions in real-world environments with real-time, dynamic, and unpredictable features.

## 6.2 EFFICIENCY ANALYSIS

We further conducted the analyses to evaluate the efficiency of our proposed RaDAgent. As making a decision step will involve a ChatGPT API call, the inefficient decision-making method would involve more API calls to accomplish the same instruction, causing costly expenses. We thus conducted experiments with varying ChatGPT API call limitations, ranging from 30 to 300, and measured the Pass Rate in ToolBench of each method under these varied limitations. The experimental results are demonstrated in Figure 1. These results showcase that the ToT-BFS, ToT-DFS, and DFSDT heavily rely on a large number of ChatGPT API calls to achieve a high Pass Rate. Once limiting the number of API calls, their performance even cannot surpass CoT. In contrast, our approach achieves the highest Pass Rate under all limitation settings, especially in low-resource settings. We attribute it to the fact that our method can utilize Elo scores to dynamically select the promising decision steps, avoiding those unpromising ones. Thus, our method illustrates superior efficiency against baselines and the practical advantages of our approach in real-world scenarios. To validate our generalized

---

[1]Note that the performance of Game of 24 is reproduced based on the `gpt-3.5-turbo-0613-16k` which is inconsistent with the reported results in the official paper.

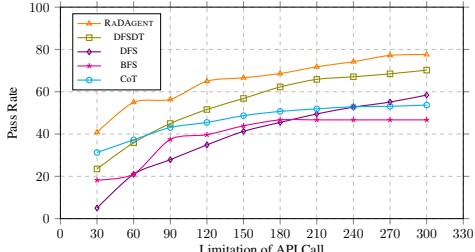
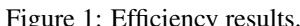

Figure 1: Efficiency results.

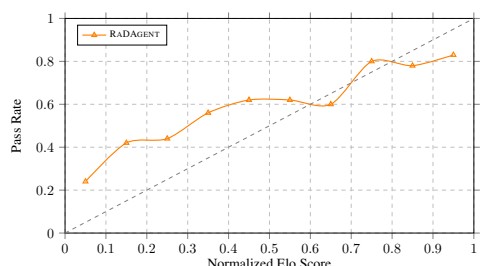

Figure 2: Performance with varied Elo scores.

efficiency advantage, We have further conducted detailed efficiency analyses on Game of 24 and the complete experimental results are presented in Appendix B.5.

### 6.3 SOLUTION RANKING

In addition to validating the effectiveness of our approach to reach feasible solutions, we seek to investigate whether RaDAgent can further provide solutions with higher quality. We adopt *Win Rate* in ToolEval from ToolBench to compare the decision sequences produced by different methods for a given instruction. Based *Win Rate*, we utilize PRP (Qin et al., 2023d) to rank decision sequences of all methods to report their rank to measure the superiority of each decision sequence. We further develop a variant of our model named *RandSelect* which selects the final decision sequence randomly while *EloSelect* selects based on the highest Elo score. We then select representative baselines (CoT@3, Reflexion, ToT-BFS, ToT-DFS, DFSDT) and conduct a comprehensive comparison of the decision

Table 3: Solution ranking experimental results on ToolBench dataset. Bold marks the best (the smaller, the better).

| Model | Pref. Rank |
|---|---|
| CoT@3 | 3.45 |
| Reflexion | 3.48 |
| ToT-BFS | 3.25 |
| DFSDT | 2.91 |
| RaDAgent | |
| w/ *RandSelect* | 3.24 |
| w/ *EloSelect* | **2.19** |

sequences produced by each method. The experimental results are summarized in Table 3, and it reveals that RaDAgent consistently achieves the top rank (2.19 on average) among all comparable baselines. Especially, *EloSelect* obviously outperforms *RandSelect*, confirming the capability of our Elo-based Utility Learning to assess the utility of each decision sequence to select superior solutions, resulting in high-quality decision-making.

### 6.4 CALIBRATION PROPERTY OF ELO-BASED UTILITY

To verify the effectiveness of our Elo-based Utility Learning in providing reliable utility assessments, we conducted a comprehensive analysis using the ToolBench dataset. As the Elo score serves as a metric to represent the utility of each decision, we seek to determine whether the Elo score is a reliable indicator of decision utility. To this end, we partitioned the ToolBench dataset into several subsets based on the Elo scores assigned to the decision sequences generated by RaDAgent. We first collected the Elo scores for all decision sequences predicted by RaDAgent in ToolBench data and then normalized them to scale within the range of 0 to 1. Next, we sorted the normalized Elo scores and divided them into 10 intervals, getting 10 subsets of ToolBench data accordingly and calculated the Pass Rate for each subset. Figure 2 illustrates the experimental results. A discernible trend is observed: the Pass Rate consistently increases with higher Elo scores. A higher Elo score indicates that the decision sequence is more likely to represent an accomplished solution to the instruction, whereas a lower Elo score suggests that the instruction may be more challenging, and the corresponding decision sequence may not effectively solve the instruction. This clear positive correlation between the Elo score and the Pass Rate demonstrates the efficacy of the Elo-based Utility Learning in providing reliable assessments of decision utility.

Table 4: Success rate of different decision measures on Game of 24.

| Method | Success Rate |
|---|---|
| DFSDT | 29.0 |
| Handcrafted Measure | 26.0 |
| LLM Measure (RaDAgent) | 43.0 |

Table 5: Comparison with MCTS variants on Game of 24.

| Method | Success Rate |
|---|---|
| MCTS@40 | 22.0 |
| MCTS@100 | 40.0 |
| RaDAgent | 43.0 |

## 6.5 RELIABILITY OF LLM EVALUATION

As we utilize LLM itself to provide decision sequence comparisons instead of handcrafted external measures, we have conducted additional experiments to validate if RaDAgent can provide more reliable measures than handcrafted external measures. We heuristically design a decision measure strategy, instead of LLM, for comparing decision sequences for Game of 24 to guide the decision-making process. The strategy is to compare two decisions to decide which result is close to 24 (i.e., the difference between the final calculation result of four numbers and 24). The closer one wins. If their difference are the same, they are tied. We replace the LLM comparison with the manual-designed strategy to conduct the experiments. The results are shown in Table 4. The performance of this handcrafted strategy (achieving only $26.0\%$ Success Rate) is notably inferior to RaDAgent even DFSDT. This handcrafted strategy does not reflect the real performance since the result of an arithmetical expression is close to 24 numerically does not mean it is close to 24 operationally. The inaccurate external measure will mislead the decision-making procedure to inferior performance.

## 6.6 RELIABILITY OF ELO RATING

As we utilize the Elo rating system to learn the internal utilities, it is essential to assess the reliability of the Elo Rating system as a utility evaluation tool. To this end, we have implemented the Monte Carlo Tree Search (MCTS) baseline with a value evaluation algorithm proposed by ToT (Yao et al., 2023) for Game of 24. Different from our method utilizing the Elo rating system, this baseline calculates the score at each decision step based on the value evaluation proposed by ToT and uses the standard *Upper Confidence Bounds for Trees* algorithm (UCT) Kocsis & Szepesvári (2006) to guide the decision-making process. We introduced two variants: MCTS@40 and MCTS@100, by conducting 40 and 100 simulations at each decision step, respectively. The outcomes of these experiments are presented in Table 5. It is evident that none of these MCTS variants could outperform RaDAgent. Despite MCTS@100 displaying a performance somewhat closer to RaDAgent, it necessitated 100 simulations for each decision step, leading to a significant number of API calls. Conversely, RaDAgent required only 20 exploratory steps to achieve superior performance, which can be attributed to the Elo scores' ability to provide precise directions for exploration guidance, which can thereby enhance both the effectiveness and efficiency of the decision-making process.

## 6.7 IMPACT OF ELO UPDATE STEP

To validate the impact of the Elo update step $K$ in the Elo rating system, we conducted a series of experiments with different values of $K = \{10, 50, 100\}$ on the Game of 24 scenario. The experimental results are listed in Table 6. Through these experiments, we observed that $K = 50$ yielded the most optimal performance for our RaDAgent. It is important to note that $K$ in the Elo update algorithm functions analogously to the learning rate in Stochastic Gradient Descent optimization algorithms (Sra et al., 2011). The choice of $K$ significantly influences the rate during Elo scores converge to their accurate values. A larger $K$ may lead to instability in the Elo scores, as it causes larger adjustments, thereby potentially overshooting the optimal value. Conversely, a smaller $K$ can result in slower convergence, necessitating more comparisons to reach an accurate assessment. Our experimental results have shown that setting $K$ as 50 can derive the best performance.

## 6.8 IMPACT OF UTILITY LEARNING PROMPT

We have conducted additional experiments to validate the impact of the prompt design in the Elo-based utility learning algorithm on the Game of 24 scenario. In this setting, we employed a more

Table 6: Success rate of different $K$ in Elo score update on Game of 24.

| Value of $K$ | Success Rate |
|---|---|
| $K = 10$ | 34.0 |
| $K = 50$ | 48.0 |
| $K = 100$ | 43.0 |

Table 7: Success rate of different prompts of the utility learning on Game of 24.

| Method | Success Rate |
|---|---|
| DFSDT | 29.0 |
| SimplePrompt | 36.0 |
| ElaboratePrompt(RaDAgent) | 43.0 |

straightforward utility learning prompt to compare two decision sequences. This prompt is detailed in Appendix A.4 and the experimental results are shown in Table 7. From the results, we can observe that there was an obvious decrease in performance with the simpler prompt compared with our original RaDAgent but it still outperforms the best baseline DFSDT. These results highlight a couple of key points: (1) Impact of Prompt Design: The experiment demonstrated that the design of the utility learning prompt does indeed impact the performance of the system. A more complex or carefully crafted prompt contributes to better utility assessment, leading to more effective decision-making. (2) Robustness of Utility Learning: Despite the reduced performance with a simpler prompt, the fact that RaDAgent continued to outperform the baseline indicates the inherent robustness of our utility learning approach. It suggests that while the prompt design is significant, the core mechanics of our Elo-based utility learning algorithm are strong enough to maintain a competitive edge even under suboptimal conditions. These findings unveil the need for further research into the optimal design of utility learning prompts. We plan to explore a broader range of prompt complexities and styles to fully understand their impact on the efficacy of the utility learning process in the future.

Additionally, to further verify the reliability of our Elo-based Utility Learning algorithm, we conduct **detailed hyperparameter analysis experiments in Appendix B** including the initial Elo score, the number of decision comparisons, initial decision solution, etc.

## 6.9 IMPACT OF LARGE LANGUAGE MODELS

To validate the effectiveness of different LLMs, we have conducted additional experiments integrating `GPT-4` into our RaDAgent instead of `GPT-3.5` on the Game of 24 scenario. The experimental results are shown in Table 8. We found that RaDAgent, leveraging the enhanced capabilities of `GPT-4`, demonstrates its superiority over its `GPT-3.5` version. This finding underscores the scalability and adaptability of RaDAgent. These results suggest the following: (1) As LLMs continue to evolve, our model can capitalize on these advancements to further enhance decision-making efficiency and accuracy. (2) Compared with the `GPT-4` version of ToT Yao et al. (2023), integrating our decision-making approach continues to yield performance improvements, demonstrating the necessity of robust decision-making strategies even when employing advanced LLMs. In summary, LLMs and decision-making approaches are complementary, mutually enhancing each other to achieve superior performance outcomes.

Table 8: Success rate of `GPT-3.5` and `GPT-4` on Game of 24. ToT is reported from its original paper (Yao et al., 2023).

| LLM | Method | Success Rate |
|---|---|---|
| GPT-3.5 | ToT | 19.0 |
| | RaDAgent | 43.0 |
| GPT-4 | ToT | 74.0 |
| | RaDAgent | 82.0 |

## 7 CONCLUSION

In this work, we have introduced RaDAgent to learn the internal utility judgment ability for agents to achieve rationality across a diverse range of real-world tasks. We design an iterative framework involving Experience Exploration and Utility Learning to enhance agents to learn numeric utility for each decision step and guide the decision-making process. Extensive experiments on the Game of 24, WebShop, ToolBench, and RestBench datasets have confirmed the effectiveness of RaDAgent, outperforming baseline methods by achieving notable improvements and producing higher-quality solutions. Moreover, the reduction in LLM API calls showcases the efficiency gains of our approach. By empowering agents with rationality, our work paves the way for their broader utilization in real-world scenarios, alleviating the reliance on external performance measures.

ACKNOWLEDGEMENTS

This work is supported by the Postdoctoral Fellowship Program of CPSF (Grant No. GZB20230343) and China Postdoctoral Science Foundation (Grant No. 2023M741945).

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

# A PROMPT DESIGN

## A.1 UTILITY LEARNING PROMPT

Our utility learning prompt is designed as follows:

```
You are value-GPT, an expert in defining which trail is better and
    closer to solving the task. Here is the task description:
*******************************
{{BEGIN_DESCRIPTION}}
your_task: {task_description}
your_query: {input_description}
{{END_DESCRIPTION}}
*******************************
Here are two candidates A and B. They both try to handle the task
   with some function calls. Their trails are as follows.
*******************************
{{CANDIDATE_A_START}}
{candidate_A}
{{CANDIDATE_A_END}}
*******************************
{{CANDIDATE_B_START}}
{candidate_B}
{{CANDIDATE_B_END}}
*******************************
```

Then, ChatGPT should call the following function[2] to give the result.

```
{
    "name": "choose_preference",
    "description": "Choose the preferred answer for the query
        within all given answers.",
    "parameters": {
        "type": "object",
        "properties": {
            "preference": {
                "type": "number",
                "description": "The index of the preferred answer
                    in all given answers."
            },
        },
    },
}
```

## A.2 DECISION-MAKING PROMPT

Our decision-making prompt is designed as follows:

```
You are the Decision-Making GPT and can perform any task using the
    tree search method.
The search method is as follows:
1. First, I will provide you with the task description and input
   details.
2. For each task, you need to call various functions through
   multiple steps.
3. At each step, you need to give your thought to analyze the
   status now and what to do next, with a function call to
   actually execute your step.
```

---

[2]https://openai.com/blog/function-calling-and-other-api-updates

```
After the call, you will get the call result, and you are now in a
    new state.
4. Each (thought-function) pair mentioned above is considered a
   tree node, and each trail is a tree path from the root to a
   terminal node. Therefore, the Monte Carlo search tree contains
    multiple trails.
5. Although you may not see previous trails, in each trail, I will
    first place you in an intermediate state determined by the "
   value" of the node, and then you make different choices from
   there.

Remember:
1. Always make a function call at each step.
2. If you believe you have gathered enough information, call the
   function "Finish: give_answer" to provide your answer for the
   task.
3. If you feel unable to handle the task from this step, call the
   function "Finish: give_up_and_restart".

Let's begin!
Task description: {task_description}
```

## A.3  TASK DESCRIPTION

The task description of Game of 24 is as follows:

```
Use numbers and basic arithmetic operations (+ - * /) to obtain
    exactly one number 24. In each step, you are only allowed to
    choose two of the left numbers to obtain a new number. For
    example, 7 * 9 - 3 * 1 3 = 2 4.
Remember:
1. All of the numbers must be used and must be used ONCE. So Only
    when the left number is exactly 24, you will win. So you don't
     succeed when the left number = [2 4, 5]. You succeed when the
     left number = [2 4].
2. All the try takes exactly 3 steps, and the task ends when the
    count of left numbers is 1, and check whether the only left
    number equals 24.
3. When there are two numbers left, ALWAYS pre-compute and list
    all the operations' combined results( + - * / ), and find if
    there is a way to combine them to 24 before you make the
    function call.
3.1. If There is a way, use function "play_24" to combine it.
3.2. If not, use function "give_up" to restart.
4. The status changes ONLY when you call function "play_24". If
    you only give thoughts, nothing happens.
5. "play_24" inputs only one step, if you want to combine many
    numbers, split it into multiple calls.
```

The task description of Webshop is listed as follows:

```
You should use functions to help handle the web shopping task on a
     webshop site.
We have 2 Pages: Product Selection Page & Product Details Page.
You have access to the following functions:
1. search: at any time, you can search a product by keywords. Then
     you will go to the Product Selection Page which shows a list
    of related products.
2. select: after searching keywords, you can select a product on
   the Product Selection Page. Then you will go to the Product
```

```
    Details Page, which shows the details of the product you
        select.
3. buy: On the Product Details Page, you can buy a product. Then
    the shopping task is completed.
```

The task description of ToolBench is listed as follows:

```
You should use functions to help handle real-time user queries.
    Remember:
1. ALWAYS call "Finish" function at the end of the task. And the
    final answer should contain enough information to show to the
    user, If you can't handle the task, or you find that function
    calls always fail(the function is not valid now), use function
     Finish->give_up_and_restart.
2. Do not use origin tool names, use only subfunctions' names.
You have access to the following tools:
```

### A.4  SIMPLE PROMPT OF UTILITY LEARNING

```
    Giving task description and candidate answers, I want you to
        choose one preferred answer which is more close to success
        .

    ******************************
    {{BEGIN_DESCRIPTION}}
    your_task: {task_description}
    your_query: {input_description}
    {{END_DESCRIPTION}}
    ******************************

    ******************************
    {{CANDIDATE_0_START}}
    {candidate_A}
    {{CANDIDATE_0_END}}
    ******************************
    {{CANDIDATE_1_START}}
    {candidate_B}
    {{CANDIDATE_1_END}}
    ******************************
```

## B  HYPERPARAMETER ANALYSIS

### B.1  SELECTION OF ELO COEFICIENT

The selection of $r$ is based on the classic Elo rating algorithm. In the classic Elo rating system (Elo & Sloan, 1978), the expected superiority is defined as:

$$E_{x>y} = \frac{1}{1 + 10^{-\frac{v_x - v_y}{400}}} \tag{10}$$

In this paper, we employ the base to compute the expected superiority for computational implementation convenience:

$$E_{x>y} = \frac{1}{1 + e^{-\frac{v_x - v_y}{r}}} \tag{11}$$

To achieve equivalence between them, we can set as approximately $r = \frac{400}{\ln 10} \approx 173.72$ to change the base. In this way, our calculation of the expected superiority equals the classic Elo rating algorithm.

## B.2 Impact of initial Elo Score

As Equation 1 shows, within the Elo rating algorithm, the decision-making process is influenced by the relative difference between Elo scores of decision steps, rather than their absolute values. The essence of the Elo rating system lies in its dynamic nature, where scores are adjusted based on pairwise comparisons over time. This means that regardless of the initial score assigned to each decision step, the subsequent adjustments made through pairwise comparisons are what determine the final, accurate assessment of each decision's utility. Therefore, the initial score primarily serves as a starting point, and its specific value is not critical to the overall decision-making process. Consequently, the initial Elo score assigned to decision steps does not fundamentally impact the outcome of the comparisons.

Table 9: Success rate of different Elo score on Game of 24.

| Value of Elo | Success Rate |
| --- | --- |
| Elo$= 0$ | 62.0 |
| Elo$= 100$ | 60.0 |
| Random Elo | 63.0 |

To validate the reliability of our Elo rating system, we further conducted two experiments. First, we manualy initialize Elo score of all decision steps as $100$ instead of $0$ in our original settings. Second, all Elo scores are initialized randomly so each decision step has different Elo scores in the begining. We re-run our method in the Game of 24 scenario and assess the final performance. The results are listed in Table 9 and it can find that different initialized Elo scores result in the similar final performance of our method. The results show that despite the different initial values, the final performance remains consistent. This consistency confirms that through multiple comparisons, Elo scores converge to their true value regardless of their initialization, thus validating the robustness of our approach. Note that we implement our method and baseline based on GPT-4o-mini to reduce the API cost so the performance does not equal those reported in the main paper.

## B.3 Impact of the Converge of Elo Rating

As the Elo score will converge after multiple comparisons, the number of comparisons may impact the final performance. In our experiments, we set the number of comparison in the utility learning as 2 per candidates, i.e., each candidate decision sequence should be compared twice. To validate the impact of the number of comparison, we double it as $4$ and re-run the experiments on Game of 24. The results are listed in Table 10 and we find that doubling the number of the comparison cannot bring improvement, revealing that Elo scores have converged to their true values. Note that we implement our method based on GPT-4o-mini to reduce the API cost so the performance does not equal those reported in the original paper.

Table 10: Success rate of different the number of comparison on Game of 24.

| Settings | Success Rate |
| --- | --- |
| #.CMP $= 2$ | 62.0 |
| #.CMP $= 4$ | 62.0 |

## B.4 Impact of Initial Decision Solution

As the utility learning are conducted after exploring multiple decision sequences for comparisons, the first explored decision sequence (i.e., initial deicision sequence) may impact the utility learning effectiveness and further influence the exploration process. To validate the robustness of our method on the initial decision sequence, we conducted additional experiments using the Game of 24 scenario. Given that the Game of 24 requires three-step decisions, we manually initialized three different decision sequences for each instance, where the first, second, and third decisions were deliberately set as the incorrect decisions. This allowed us to test our method's resilience across different initial

Table 11: Success rate of different initial decision solution on Game of 24.

| Settings | Success Rate |
|---|---|
| Init. w/ Incorrect First Step | 59.0 |
| Init. w/ Incorrect Second Step | 61.0 |
| Init. w/ Incorrect Third Step | 60.0 |

conditions. The experiment results are listed in Table 11, Despite the different initial decision sequences, our method consistently achieved similar success rates, demonstrating its robustness. Note that we implement our method and baseline based on GPT-4o-mini to reduce the API cost so the performance does not equal those reported in the main paper.

The Elo rating system is specifically designed to be resilient (Elo, 1967), even when initial decision policies are suboptimal. This is achieved by continuously comparing and updating the Elo scores accordingly, allowing the Elo scores to progressively converge to their true values. Adapting the Elo rating system in our method, it can provide reliable measure the utility of each decision step. Even if the initial decision sequence is suboptimal, through iteratively experience exploration and utility learning, each decision step would be measured accurately and our method would avoid the suboptimal and even poor decisions. As a result, the effectiveness of our method does not heavily depend on the quality of the initial decision sequences. Instead, it is capable of learning and improving from its experiences, adapting to better strategies as it gains more insights through repeated interactions. These results further confirm that our method can effectively adapt and converge to optimal decision paths, even when starting from suboptimal sequences.

## B.5 EFFICIENCY ANALYSIS ON GAME OF 24

To further validate the generalized efficiency of our method, we further conducted efficiency experiments in the Game of 24 scenario. Similar to the settings in § 6.2, we manualy set different API call budgets and assess the final performance of our method against the best baseline, DFSDT. The results are listed in Table 12. Obviously, we can find that the similar efficiency results show in the Game of 24. Our method still achieve Highest Performance for Same Cost and Lowest Cost for Same Performance. Such results further validates the efficiency superiority of our method against baselines. Note that we implement our method and baseline based on GPT-4o-mini to reduce the API cost so the performance does not equal those reported in the main paper.

Table 12: Efficiency analysis of different methods on Game of 24.

| Model | API call budget | | | |
|---|---|---|---|---|
| | 50 | 100 | 150 | 200 |
| DFSDT | 16.0 | 32.0 | 35.0 | 42.0 |
| RaDAgent | 20.0 | 38.0 | 52.0 | 55.0 |

## B.6 GENERALIZABILITY TO VARIOUS LARGE LANGUAGE MODELS

To verify the generalizability of RaDAgent to various LLMs including open-source LLMs, we have conducted additional experiments using two alternative representative LLMs: the proprietary `claude-3-5-haiku-20241022` and the open-source `llama-3.1-8b`. We re-implemented RaDAgent with these models and evaluated its performance on the Game of 24 task. The experimental results are presented in the Table 13. The results demonstrate that RaDAgent continues to yield significant performance improvements when integrated with different LLMs, underscoring the robustness and generalizability of our proposed approach.

Table 13: Performance based on various LLMs on Game of 24.

| LLM | CoT@1 | RaDAgent |
|-----|-------|----------|
| llama-3.1-8b | 3.0 | 15.0 |
| claude-3-5-haiku | 2.0 | 71.0 |
| gpt-3.5-turbo | 6.0 | 43.0 |

## B.7 ERROR ANALYSIS

In this section, we present comprehensive analysis to show the failure in decision-making in Tool-Bench. We commence our analysis by categorizing the common reasons for failure encountered by each model in ToolBench. These reasons encompass: (1) **Tool Inaccessibility**: Occurrences where a subset of the designated tools is inaccessible, e.g., HTTP 404 or 500 error. (2) **Parameter Error**: Occurrences when call tools, including parameter format mismatching and missing mandatory parameter fields. (3) **Tool Hallucination**: Instances where the model employs tools not provided, i.e., invoking a non-existent tool. (4) **Decision Failure**: Instances where the model fails to accomplish although none of the aforementioned problems occur. We present the incidence ratio of the aforementioned categories. Specifically, the incidence ratios are calculated based on the entire exploration process. If the error occurs in the explored decision tree (even not in the final decision sequence), it is still counted. As Tool Hallucination and Parameter Error can be fixed during decision making, we further report the fix ratio that agents successfully accomplish the instructions although errors occurred.

Table 14: Incidence ratio and Fix ratio of common failure reasons in ToolBench dataset.

| Method | Tool Hallucination | | Parameter Error | | Tool Inaccessibility | Decision Failure |
|--------|-----------|------|-----------|------|----------------------|------------------|
| | Incidence | Fix | Incidence | Fix | | |
| CoT@3 | 14.2 | 25.4 | 41.2 | 14.8 | 2.0 | 52.5 |
| BFS | 18.8 | 25.5 | 50.8 | 31.1 | 2.6 | 48.6 |
| DFSDT | 31.5 | 38.9 | 62.5 | 41.0 | 3.0 | 26.4 |
| RaDAgent | 42.1 | **53.3** | 62.3 | **54.0** | 3.0 | **14.8** |

From Table 14, several noteworthy observations arise: (1) RaDAgent boasts the lowest incidence ratio of decision failure, highlighting its adeptness in decision making. (2) As RaDAgent conducts a diverse and extensive exploration, it will experience more parameter errors and tool hallucinations, causing a higher incidence ratio. This diverse exploration is integral as it allows RaDAgent to thoroughly evaluate a wide array of possible decision pathways, even those that are less conventional or more prone to errors. Despite exploring riskier or more error-prone decisions, RaDAgent effectively utilizes the Elo-based utility learning mechanism to learn from diverse explorations and subsequently pinpoint the most efficient and error-free pathway. The high fix ratio underlines RaDAgent's ability to rectify potential errors encountered during the exploration phase, ultimately leading to a reliable and effective decision-making process. (3) All methods own similar incident ratio of tool inaccessibility which shows that there still exist some inoperative APIs in ToolBench, influencing the decision-making process. (4) We examine cases that all methods fail and find certain cases remain unsolvable due to the ambiguity of user-provided values (e.g., user ID, email address) or restrictions imposed by limited tool chain lengths, which underscores the necessity for advanced decision-making proficiencies.

## C BROADER IMPACT

This paper presents work whose goal is to advance the field of Autonomous Decision-Making for Large Language Models. There are many potential broader impacts of our work and we discuss some aspects in the following: Firstly, our work explores the internal rationality for LLM-based agents, and the proposed Elo-based Utility Learning could leverage Elo Rating system to construct quantitative utilities for each decisions. This advancement could have profound implications for industries where decision-making is critical, such as finance, healthcare, and law. However, this autonomy raises ethical considerations regarding the extent to which LLMs should be allowed to make decisions

without human oversight, especially in high-stakes scenarios. Secondly, as RaDAgent demonstrates improved efficiency and cost-effectiveness in decision-making tasks, there could be an increased reliance on LLMs in various sectors. This dependence may lead to a reduction in human labor for certain tasks and could influence the job market, necessitating a re-evaluation of workforce skills and training. Thirdly, while RaDAgent learns from its posterior experiences, the quality and diversity of these experiences are crucial. There is a potential risk of inheriting biases present in the training data or developing new biases based on limited or skewed experiences. This aspect necessitates continuous monitoring and updating of the model to ensure fairness and impartiality in decision-making.

## D    LIMITATIONS

Our approach still has several limitations: (1) Our method involves exploring new decision traces from intermediate decision steps, necessitating the recovery of the state at each step. In practice, certain decisions cannot be reversed once executed. In these instances, our method requires a sophisticated rollback mechanism to function correctly. (2) Our utility learning method relies on the comparative judgment capabilities of large language models (LLMs) to achieve Elo ratings. While GPT-3.5 and GPT-4 can implement our method, it is uncertain if other LLMs, especially for open-source LLMs, can achieve similar performance. We will explore these limitations in the future.

