# OpenReview forum: "Rational Decision-Making Agent with Learning Internal Utility Judgment"
_ICLR.cc/2025/Conference — ICLR 2025 Poster_

### Official Review · Reviewer_v6Tk · 2024-10-28

**Soundness:** 2
**Presentation:** 3
**Contribution:** 3
**Rating:** 6
**Confidence:** 3

**Summary:**

This work proposes RaDAgent (Rational Decision-Making Agent) that utilizes experience exploration and utility learning. Notably, Elo scores are assigned to each step to evaluate its utilities, which helps the agent to derive the optimal decisions. The authors provide empirical evidence that validates the efficacy of the proposed method on several tasks such as Game of 24, WebShop, and ToolBench. They also show that LLMs and the decision-making approaches are complementary, mutually enhancing each other to achieve superior performance outcomes.

**Strengths:**

Overall, the idea of this work is novel and intuitive.

Overall, this paper is well-written and well-presented.

The proposed RaDAgent method significantly outperforms the baseline methods across most of tasks.

**Weaknesses:**

I would recommend putting the Related Work section after the Introduction so that readers can gain more background information of this work. Indeed, explaining how the literature or baselines work can help the readers better understand the motivation of authors' idea.

The authors need to enhance the language and fix typos/grammar issues, e.g., in line 96-97, "take action", line 139 "Elo score" should be plural, etc.

Line 97-98, in general, one would use different functions to represent the policy and transition functions in MDP. I would expect a better introduction of MDP on page 2. For example, one can use a tuple (S, A, R, P, \mu) to represent an MDP. Please see more details in RL paper or Sutton's RL book. In addition, \pi is often used to denote the policy (action generator).

Even though mentioning in the Limitation section by authors, I am still interested in if RaDAgent can be implemented in other (at least one state-of-the-art) LLMs (e.g., Claude, Llama), especially not the GPT-series. Also, why the authors choose GPT-series as their backbone LLM? If RaDAgent does not work for other LLMs like Claude, what could be the possible reasons?

**Questions:**

Please refer to the "weakness" section.

---

> ### Author Response · Authors · 2024-11-24
> **Response to Related Work Section**
>
> Q: I would recommend putting the Related Work section after the Introduction so that readers can gain more background information of  this work. Indeed, explaining how the literature or baselines work can  help the readers better understand the motivation of authors' idea.
>
> A: Thank you for your constructive feedback. We appreciate your suggestion to move the Related Work section after the Introduction Section to provide better background information and enhance the understanding of our work in our revised paper.

---

> > ### Comment · Reviewer_v6Tk · 2024-11-26
> > **No revised paper**
> >
> > Thank you for your reply. However, I didn't see the revised version of paper that addressed my concern.

---

> ### Author Response · Authors · 2024-11-24
> **Response to Typos Issues**
>
> Q: The authors need to enhance the language and fix typos/grammar issues,  e.g., in line 96-97, "take action", line 139 "Elo score" should be  plural, etc.
>
> A: Thank you for pointing out these language and grammatical issues. We greatly appreciate your careful review and constructive feedback. We have conducted a thorough review of the manuscript to improve the writing quality, correct grammatical mistakes, and enhance overall readability.

---

> ### Author Response · Authors · 2024-11-24
> **Response to MDP Notation**
>
> Q: Line 97-98, in general, one would use different functions to represent  the policy and transition functions in MDP. I would expect a better  introduction of MDP on page 2. For example, one can use a tuple (S, A,  R, P, \mu) to represent an MDP. Please see more details in RL paper or  Sutton's RL book. In addition, \pi is often used to denote the policy  (action generator).
>
> A: Thank you for highlighting the inconsistency in the symbols and the need for a more rigorous introduction to MDP. We greatly appreciate your attention to detail and constructive feedback. In response to your comment, we have revised the Preliminaries section to provide a clearer and more formal introduction to MDPs. Specifically, we now define MDPs using the standard tuple notation as suggested, and have ensured consistency in the use of symbols throughout the paper. Thank you again for your valuable input.

---

> > ### Comment · Reviewer_v6Tk · 2024-11-26
> > **Can't find the revised paper**
> >
> > Unfortunately, I couldn't find the revised paper. Also, please color your modifications so that we can easily see what part the authors changed.

---

> > > ### Author Response · Authors · 2024-11-27
> > > **Revised paper has been submitted**
> > >
> > > Thank you for your valuable suggestions. We have submitted the revised version of our paper to the OpenReview system. In this revision, the Preliminaries Section has been placed immediately after the Introduction section. For details, please refer to Page 3, Lines 124–161. Additionally, all corrections related to typographical errors are highlighted in blue, while modifications to the notation of the MDP are marked in red.

---

> ### Author Response · Authors · 2024-11-24
> **Response to Re-implementation with Other LLMs**
>
> Q: Even though mentioning in the Limitation section by authors, I am still  interested in if RaDAgent can be implemented in other (at least one  state-of-the-art) LLMs (e.g., Claude, Llama), especially not the  GPT-series. Also, why the authors choose GPT-series as their backbone  LLM? If RaDAgent does not work for other LLMs like Claude, what could be  the possible reasons?
>
> A: We appreciate the reviewer’s insightful question regarding the generalizability of RaDAgent to non-GPT-series LLMs. To address this concern, we have conducted additional experiments using two alternative representative LLMs: the proprietary `claude-3-5-haiku-20241022` and the open-source `llama-3.1-8b`. We re-implemented RaDAgent with these models and evaluated its performance on the Game of 24 task. The experimental results are presented in the table below:
>
> | Model            | CoT@1 | RaDAgent |
> |-------------------|-------|----------|
> | llama-3.1-8b     | 3.0     | 15.0       |
> | claude-3-5-haiku | 2.0     | 71.0       |
> | gpt-3.5-turbo    | 6.0     | 43.0       |
>
> The results demonstrate that RaDAgent continues to yield significant performance improvements when integrated with different LLMs, underscoring the robustness and generalizability of our proposed approach.
>
> We chose the GPT-series as the backbone model in our original submission because it is currently one of the most representative and widely used LLMs, with strong performance across diverse tasks. Its representativeness as a benchmark model makes it an ideal choice for validating new methods. However, the successful replication of results with Claude and Llama further highlights that **RaDAgent is not restricted to a specific LLM family and is robust and generalized to various LLMs**.
>
> We hope this additional analysis adequately addresses the reviewer’s concerns and further supports the validity of our method.

---

> > ### Comment · Reviewer_v6Tk · 2024-11-26
> > **LLMs**
> >
> > Thank you for your additional experiments, especially given this short time of rebuttal. It is very interesting that Claude did a very good job while Llama didn't. I also noticed that the authors only compare with one baseline model in one game. I fully understand that there is not too much time in the rebuttal. However, I strongly encourage the authors to complete the experiments for Claude and Llama as in Table 1 in their camera-ready version of paper. Overall, the new experiments indeed addressed part of my concerns.

---

> > > ### Author Response · Authors · 2024-11-27
> > > **Response to adding the new experiments in camera-ready paper**
> > >
> > > Thanks for your suggestions. We understand your suggestion regarding the inclusion of comprehensive experiments for Claude and Llama in the camera-ready version of paper. We will carefully complete and analyze the results from these new experiments and ensure they are included in the camera-ready version of the paper to provide a more complete evaluation. Thank you again for your encouragement and valuable suggestions.

---

> > > > ### Comment · Reviewer_v6Tk · 2024-11-27
> > > > **Reply**
> > > >
> > > > Thank you for providing the revised paper. I will maintain my positive score.

---

### Official Review · Reviewer_edTJ · 2024-10-31

**Soundness:** 3
**Presentation:** 3
**Contribution:** 3
**Rating:** 6
**Confidence:** 3

**Summary:**

This paper introduces a novel approach named RaDAgent that addresses the limitation of LLM-based agents' dependence on external performance measures for decision-making. Unlike existing methods that rely on potentially unreliable or unavailable external metrics, RaDAgent develops internal rationality through an iterative framework combining Experience Exploration and Utility Learning. Besides, the framework incorporates the Elo Rating system to learn and assign utilities to decision steps through pairwise comparisons, enabling agents to make autonomous decisions based on learned internal judgment.

**Strengths:**

The main strength of this paper are:
1. The paper proposes a novel idea by shifting away from the conventional reliance on external performance measures to an internal utility judgment approach for LLM-based agents.
2. The framework provides a practical solution for developing internal rationality in LLM-based agents.
3. The experimental results demonstrate promising performance across multiple datasets,

**Weaknesses:**

The  main weakness of this paper are:
1. The Elo rating system might oversimplify the complexity of decision utilities. Besides, the pairwise comparison approach could be time-consuming for complex tasks.
2. The convergence for the Elo-based utility learning lacks theoretical analysis. While the paper demonstrates empirical effectiveness of the Elo rating system in learning decision utilities, it fails to provide rigorous theoretical guarantees about the convergence properties of this approach. An analysis of whether and how quickly the Elo scores converge to stable values would strengthen the theoretical foundation of the approach and provide better understanding of its reliability in different scenarios.

**Questions:**

Since the paper lacks convergence analysis of this approach, I wonder how much additional computational overhead this method requires compared to existing approaches. The time cost of conducting multiple comparisons and updating Elo scores, especially for complex decision scenarios, could be substantial and needs to be evaluated against baseline methods.

---

> ### Author Response · Authors · 2024-11-24
> **Response to the Computational Complexity of RaDAgent**
>
> Q: the pairwise comparison approach could be time-consuming for complex tasks. I wonder how much additional computational overhead this method requires compared to existing approaches. The time cost of conducting multiple comparisons and updating Elo scores, especially for complex decision scenarios, could be substantial and needs to be evaluated against  baseline methods.
>
> A:  We acknowledge that advanced decision-making algorithms (e.g., DFS,  BFS), including our method, can incur higher computational costs. Hence,  in our paper, we have conducted efficiency experiments (Section 5.3) to  evaluate performance under the same computational budget. As detailed  in Line 358-369, we compared our algorithm with various methods using a  fixed maximum number of API calls. We give part of the experimental results in the following table (details can refer to Figure 1 in the paper):
>
> | API Call Limitation | 30   | 60   | 90   | 120  | 150  | 180  | 210  | 240  | 270  | 300  |
> |------|------|------|------|------|------|------|------|------|------|------|
> | CoT  | 31.3 | 37.3 | 43.1 | 45.5 | 48.7 | 50.7 | 51.9 | 52.9 | 53.1 | 53.7 |
> | BFS  | 18.2 | 21.0 | 37.5 | 39.7 | 43.9 | 46.7 | 46.7 | 46.7 | 46.7 | 46.7 |
> | DFS  | 5.0  | 21.0 | 27.9 | 34.9 | 41.3 | 45.5 | 49.5 | 52.7 | 55.1 | 58.4 |
> | DFSDT| 23.5 | 35.9 | 45.0 | 51.6 | 56.8 | 62.3 | 65.9 | 67.1 | 68.5 | 70.2 |
> | Ours | 40.8 | 55.0 | 56.4 | 65.0 | 66.6 | 68.6 | 71.8 | 74.2 | 77.2 | 77.6 |
>
> The results from our experiments  demonstrate that:
> 1. Highest Performance for Same Cost: Our method consistently achieves the highest performance across all API budget constraints.
> 2. Lowest Cost for Same Performance: When matched for the same  performance, our method incurs the lowest computational cost compared to  other methods.  This efficiency is achieved through the iterative refinement of  decision-making, which allows our method to focus on more promising  decision paths and reduce unnecessary computations.
>
> To further validate the efficiency of our method, we further  conducted efficiency experiments in the Game of 24 scenario and this experiment is reported in Appendix B.5. Similar to  the settings in Section 5.3, we manualy set different API call budgets  and assess the final performance of our method against the best  baseline, DFSDT. The results are listed in the following table.  Obviously, we can find that the similar efficiency results show in the  Game of 24. Our method still achieves Highest Performance for Same Cost  and Lowest Cost for Same Performance. Such results further validates the  efficiency superiority of our method against baselines.
>
> | API Call Limitation | 50   | 100  | 150  | 200  |
> |------------------|-------|-------|-------|-------|
> | DFSDT           | 16.0  | 32.0  | 35.0  | 42.0  |
> | Ours            | 20.0  | 38.0  | 52.0  | 55.0  |
>
> We appreciate the reviewer’s concern regarding the potential computational overhead of the pairwise comparison approach, particularly for complex tasks. To address this, we conducted an analysis of RaDAgent's API call usage during the Game of 24 task. On average, RaDAgent requires 209.74 API calls to complete one Game of 24 task, with 82.04 API calls used for decision-making and 127.7 API calls allocated for solution comparisons to calculate Elo scores.
>
> | Task                 | Average API Calls | Percentage (%) |
> |----------------------|--------------------|----------------|
> | Decision-making      | 82.04             | 39.1%          |
> | Elo comparison  | 127.7             | 60.9%          |
> | Total            | 209.74        | 100%      |
>
>
> Notably, the API calls for comparisons constitute the majority of the total API calls, while decision-making accounts for a smaller proportion. This highlights the efficiency of RaDAgent's decision process, where **the use of Elo score calculations significantly reduces the number of API calls needed for decision-making**. As a result, the overall API call count is lower than that of baseline methods to achieve similar performance.
>
> This analysis demonstrates that while solution comparisons require additional API calls, the efficiency gained in decision-making offsets this overhead, ensuring that the method remains computationally efficient overall. We hope this addresses the reviewer’s concerns and provides clarity on the computational trade-offs of our approach.

---

> ### Author Response · Authors · 2024-11-25
> **Response to the Selection of Elo Rating System**
>
> We would like to clarify that our choice of the Elo rating system was not made to simplify the problem but rather as the result of careful consideration. Elo’s suitability for learning internal utility judgment stems from the following three key characteristics:
> 1. Dynamic Nature: Our goal is to enable the agent to make decisions independently of external metrics, requiring the agent to construct an internal utility judgment during dynamic exploration. This exploration process involves dynamically generating various decision-making sequences. The Elo algorithm supports the evaluation of dynamically generated comparison candidates, making it particularly suitable for measuring the utility of each decision within a dynamic decision-making process.
> 2. Convergence: For the agent to make optimal decisions, it needs an accurate internal utility judgment. Any bias in utility estimation could negatively impact the final performance. Elo’s convergence property ensures that as the number of comparisons increases, Elo scores converge to true utility values, thereby guaranteeing the effectiveness of the agent’s final decisions.
> 3. Quantification: During exploration, the agent needs to select superior candidates from multiple decision options to continue exploring further. The Elo algorithm provides a numerical representation of decision quality, enabling a more precise and controllable exploration process.
>
> Given these considerations, we selected the Elo algorithm to facilitate the agent's ability to learn internal utility judgment. Furthermore, we conducted calibration experiments (refer to Section 5.4), which demonstrated a strong alignment between Elo scores and task performance. Specifically, higher Elo scores consistently correspond to better performance, validating the reliability of Elo scores in reflecting decision quality. And extensive experimental results on Game of 24, Webshop, ToolBench, RestBench significantly validate the effectiveness of our RaDAgent in decision-making.
>
>
> | Pass Rate | Elo Score   |
> |--------------|-------|
> | 0.05         | 0.24  |
> | 0.15         | 0.42  |
> | 0.25         | 0.44  |
> | 0.35         | 0.56  |
> | 0.45         | 0.62  |
> | 0.55         | 0.62  |
> | 0.65         | 0.6   |
> | 0.75         | 0.8   |
> | 0.85         | 0.78  |
> | 0.95         | 0.83  |

---

> > ### Comment · Reviewer_edTJ · 2024-11-25
> >
> > Thanks for your response ! I will keep my score.

---

### Official Review · Reviewer_CTwe · 2024-11-03

**Soundness:** 2
**Presentation:** 2
**Contribution:** 2
**Rating:** 5
**Confidence:** 2

**Summary:**

The paper studies how to improve the performance of LLM-agents on sequential decision making tasks. It discussed the drawbacks of relying on pre-defined external metrics to guide the decision making of LLM-agents. Furthermore, they propose to use Elo score together with Experience Exploration techniques.

**Strengths:**

The proposal of using Elo score is reasonable and points out an interesting direction of rethinking how evaluation metrics can guide the decision making of LLM agents.

Meanwhile, the experiments are good and extensive.

**Weaknesses:**

1) the paper is rather empirical and heuristic, with combination of two heuristic techniques.
2) Elo itself also has significant issues [1]. For examples, it may provide no useful signals sometimes. In other words, such a black-box use of Elo without significant extensions/more detailed discussions on its potential limitations is a sign of limited novelty.

[1] Re-evaluating evaluations. NeurIPS 2018

**Questions:**

See weakness

---

> ### Author Response · Authors · 2024-11-28
> **Response to the "the paper is empirical and heuristic" (Part 1)**
>
> Thank you for your feedback. Below, we provide detailed responses to address the concerns raised.
>
> 1. **Over-reliance on External Metrics is A Critical Bottleneck**
>
> The reliance on external metrics is a significant limitation for current agent systems.
>
> External metrics, while widely used, are often **inherently biased (e,g., flawed or even erroneous)** and may fail to accurately capture the true quality of decisions in agent scenario [1,2,3,4]. This bias arises from their design, which is typically tailored to specific tasks or limited scenarios. Consequently, agents relying on such metrics can produce suboptimal decisions, as their performance is constrained by the inaccuracies or limitations of the chosen metric. **This issue fundamentally caps the agent's potential effectiveness, as its decision-making process is tied to the upper bound defined by the external metric's precision.**
>
> In addition to being biased, external metrics are also task-specific and **hard to generalize across diverse applications**. Each metric requires careful customization for a particular task, which limits the agent's ability to adapt to new or unseen scenarios. As highlighted in OpenAI o1 [5], current agents that heavily depend on such metrics may perform well in specialized domains like math, coding, or QA but exhibit significant limitations when faced with broader tasks. **This lack of generalization severely restricts their utility and adaptability in real-world environments.**
>
> Addressing this dependency is crucial to unlocking agents’ broader utility. Our work aims to reduce this reliance by enabling agents to develop internal utility judgments, allowing them to dynamically adapt to diverse tasks without needing predefined external metrics.
>
> ---
>
> [1] Shunyu Yao, et al. Tree of thoughts: Deliberate problem solving with large language models. NeurIPS 2023.
>
> [2] Shibo Hao, et al. Reasoning with language model is planning with world model. EMNLP 2023.
>
> [3] Bilgehan Sel, et al. Algorithm of thoughts: Enhancing exploration of ideas in large language models. ICML 2024.
>
> [4] Weijie Lv, et al. Codeact: Code adaptive compute-efficient tuning framework for code llms. arXiv 2024.
>
> [5] OpenAI. Learning to Reason with LLMs. https://openai.com/index/learning-to-reason-with-llms/

---

> ### Author Response · Authors · 2024-11-28
> **Response to the "the paper is empirical and heuristic" (Part 2)**
>
> 2. **Theoretical Foundation: Elo Algorithm for Internal Utility Learning**
>
> To address the limitations of external metrics, our method introduces internal utility learning based on posterior exploration experiences. This involves constructing a stable and reliable utility function during dynamic exploration. To achieve this, we select the Elo rating system based on careful consideration. Elo’s suitability for learning internal utility judgment stems from the following three key characteristics:
> - **Dynamic Nature**: Our goal is to enable the agent to make decisions independently of external metrics, requiring the agent to construct an internal utility judgment during dynamic exploration. This exploration process involves dynamically generating various decision-making sequences. The Elo algorithm supports the evaluation of dynamically generated comparison candidates, making it particularly suitable for measuring the utility of each decision within a dynamic decision-making process.
> - **Convergence**: For the agent to make optimal decisions, it needs an accurate internal utility judgment. Any bias in utility estimation could negatively impact the final performance. Elo’s convergence property ensures that as the number of comparisons increases, Elo scores converge to true utility values, thereby guaranteeing the effectiveness of the agent’s final decisions.
> - **Quantification**: During exploration, the agent needs to select superior candidates from multiple decision options to continue exploring further. The Elo algorithm provides a numerical representation of decision quality, enabling a more precise and controllable exploration process.
>
> Given these considerations, we selected the Elo algorithm to facilitate the agent's ability to learn internal utility judgment. Extensive theoretical studies validate the robustness and reliability of the Elo algorithm such as chess [1,2], athletic sports [3-5], video games [6-7], biometrics [8], LLM evaluation [9], etc. Moreover, our calibration experiments (Section 5.4) demonstrate a strong correlation between Elo scores and task performance, further supporting the validity of our choice. These results underscore Elo’s suitability for learning internal utility judgments in dynamic decision-making processes.
>
> ---
>
> [1] AE Elo. The proposed uscf rating system, its development, theory, and applications. chess life xxii (8): 242–247, 1967.
>
> [2] Backgammon Ratings Explained. results.ukbgf.com.
>
> [3] Lyons, Keith. What are the World Football Elo Ratings?. The Conversation.
>
> [4] Postseason Odds, ELO version. Baseballprospectus.com.
>
> [5] Cole, Bryan. Elo rankings for international baseball. Beyond the Box Score. SB Nation.
>
> [6] Matchmaking | LoL – League of Legends. Na.leagueoflegends.com.
>
> [7] AirMech developer explains why they use Elo. https://www.carbongames.com/forums/viewtopic.php?p=83043#p83043
>
> [8] Using Comparative Human Descriptions for Soft Biometrics. International Joint Conference on Biometrics (IJCB), 2011
>
> [9] Chatbot Arena Leaderboard Week 8: Introducing MT-Bench and Vicuna-33B | LMSYS Org. lmsys.org.

---

> ### Author Response · Authors · 2024-11-28
> **Response to the "the paper is empirical and heuristic" (Part 3)**
>
> 3. **Novel Contributions Beyond Integrating Elo into Agent Systems**
>
> Our approach is not a straightforward application of Elo but incorporates innovative mechanisms to address critical challenges in integrating Elo into agent decision-making:
> - Reliable Elo Score Computation for Decision Sequences: Traditional Elo evaluates static comparisons between two players. However, in our context, the comparisons involve two series of decision sequences, where intermediate steps lack final environment feedback. To mitigate this, we compare complete decision sequences (including final feedback) and propagate scores backward from leaf nodes (see Equation 7 in Line 247). This ensures that intermediate scores are accurate and credible.
> - Elo-Driven Decision-Making with Efficiency and Exploration: Elo scores are inherently less reliable with limited comparisons. Directly using them for decision-making risks suboptimal exploration. To address this, we adopt a softmax-based random sampling strategy with rejection:
>   - Randomization avoids premature exploitation of non-global optima before Elo scores converge.
>   - Rejection sampling preserves the agent’s ability to explore unvisited decisions, ensuring comprehensive exploration.
>
> These innovations go beyond the standard Elo algorithm and are specifically designed to enhance agent decision-making in dynamic environments.

---

> ### Author Response · Authors · 2024-11-28
> **Response to the "the paper is empirical and heuristic" (Part 4)**
>
> 4. **Robustness and Generalization: Real-World Validation**
>
> Our proposed method has been proven its value in the real-world scenarios. Specifically, our research primarily focuses on  real-world datasets, rather than simulated environments to ensure that  our method is robust and applicable to authentic scenarios:
> - ToolBench Dataset is sourced from APIs available on the real-world  API Hub platform, RapidAPI. It includes APIs from widely-used services  such as Google and Twitter, reflecting actual application scenarios.  This demonstrates the practical utility of our approach in environments  where these APIs are actively used.
> - WebShop Dataset contains content cached from the actual Amazon  website, providing a realistic e-commerce environment for testing. By  using this dataset, we validate our method's performance in dynamic and  complex commercial settings, akin to real-world shopping scenarios.
> - RestBench  dataset features Restful APIs from two prominent real-world  applications: TMDB and Spotify. These datasets include human-annotated  real-world tasks with ground truth decision sequences.
>
> By comparing our  method to baseline methods, we have  demonstrated that our approach outperforms the baselines and achieves  the best performance. This result underscores the robustness and generalization of  our method in making decisions in authentic real-world environments.
>
> We appreciate the reviewers’ suggestions and welcome additional feedback to further enhance our work. If there are specific areas where more theoretical analysis or alternative approaches could strengthen the paper, we are open to incorporating these improvements.

---

> ### Author Response · Authors · 2024-12-02
> **Response to Elo issues (Part 1)**
>
> First, we would like to calrify that as we said in the Response to "the paper is empirical and heuristic" (Part 2), we **select the Elo rating system based on careful consideration**. Elo’s suitability stems from three key characteristics:
> - **Dynamic Nature**: Our goal is to enable the agent to make decisions independently of external metrics, requiring the agent to construct an internal utility judgment during dynamic exploration. This exploration process involves dynamically generating various decision-making sequences. The Elo algorithm supports the evaluation of dynamically generated comparison candidates, making it particularly suitable for measuring the utility of each decision within a dynamic decision-making process.
> - **Convergence**: For the agent to make optimal decisions, it needs an accurate internal utility judgment. Any bias in utility estimation could negatively impact the final performance. Elo’s convergence property ensures that as the number of comparisons increases, Elo scores converge to true utility values, thereby guaranteeing the effectiveness of the agent’s final decisions.
> - **Quantification**: During exploration, the agent needs to select superior candidates from multiple decision options to continue exploring further. The Elo algorithm provides a numerical representation of decision quality, enabling a more precise and controllable exploration process.

---

> ### Author Response · Authors · 2024-12-02
> **Response to Elo issues (Part 2)**
>
> Second, we **not** just simply utilize Elo rating algorithm in a black-box manner without significant extensions/more detailed discussions on its potential limitations. As we said in Response to the "the paper is empirical and heuristic" (Part 3), **vallina Elo rating algorithm cannot be adopted in agent decision making process directly and we have designed two techniques to modify the calculation mechanism of Elo rating algorithm**: bottom-up style Elo score calculation for intermediate decision steps and Elo-dirven decision exploration. Details can be found in Rebuttal (Response to the "the paper is empirical and heuristic" (Part 3) and Section 4.1 and 4.2 in our paper). These techniques are proposed beyond vallina Elo rating algorithm as the novelty of our method.

---

> ### Author Response · Authors · 2024-12-02
> **Response to Elo issues (Part 3)**
>
> Finally, about your mentioned the issues of Elo rating algorithm, in fact, **we have carefully considered its applicability and mitigated its potential shortcomings based on prior research[1,2,3,4]**. Below, we provide a detailed response to the issues raised:
>   - **Inability to Handle Cyclic Games**: As acknowledged, Elo is unsuitable for cyclic game scenarios, such as rock-paper-scissors, where no clear transitive hierarchy exists. However, in the context of decision-making tasks, cyclic dependencies do not arise. The quality of a decision sequence can be objectively and unambiguously assessed based on its success rate in completing the final task. This success rate inherently provides a monotonic and order-preserving measure of performance, ensuring that all decision sequences can be meaningfully ranked without cyclic ambiguities.
>   - **Inapplicability to Team Competitions**: Elo is designed for individual matchups and does not natively support team-based competition formats. However, this limitation is irrelevant to our setting, as decision-making tasks do not involve team-versus-team competitions. Each decision sequence operates independently, and its quality is evaluated in isolation against other sequences. Thus, this limitation has no impact on our methodology.
>    - **Lack of Adaptation to Temporal Changes in Skill**: Elo scores may fail to capture the evolution of a participant’s skill over time. However, this limitation is also irrelevant to our setting. The quality of a decision sequence in our context is static and determined solely by its intrinsic design and effectiveness, which do not evolve over time. Therefore, this limitation does not affect our methodology.
>   - **Impact of Imbalanced Comparisons**: The accuracy of Elo scores can be influenced by imbalanced match comparisons, such as scenarios where some participants engage in fewer matches or compete primarily against low-ranked opponents. In our work, we explicitly address this concern in `Line 235` of the our paper, where we describe the use of random sampling to ensure fair and balanced comparisons between decision sequences. This approach minimizes the risk of imbalances and ensures that decision sequences are evenly compared across a diverse range of Elo scores.
>
>
> Additionally, we acknowledge the existence of extensions to the Elo algorithm, such as TrueSkill [1,2], Glicko [3,4] including your mentioned paper [5], addressing some of these limitations. However, given the absence of cyclic games, team dynamics, or temporal variability in decision-making tasks, we determined that the standard Elo framework is well-suited for our needs. Furthermore, we have adapted the internal calculation of the Elo algorithm to better align with the requirements of our specific scenario, ensuring its robust applicability rather than employing it as a “black-box” solution.
>
> ---
>
> [1] Herbrich, Ralf et al. TrueSkill : A Bayesian Skill Rating System. NeurIPS 2019.
>
> [2] Minka, Tom et al. TrueSkill 2: An improved Bayesian skill rating system. https://www.microsoft.com/en-us/research/publication/trueskill-2-improved-bayesian-skill-rating-system/
>
> [3] Glickman, Mark. The Glicko System. http://www.glicko.net/glicko/glicko.pdf
>
> [4] Professor Glickman's Glicko-Website. http://www.glicko.net/glicko.html
>
> [5] Re-evaluating evaluations. NeurIPS 2018

---

> ### Author Response · Authors · 2024-12-02
> **Expect Your Feedback Before the Rebuttal Deadline**
>
> Dear Reviewer CTwe,
>
> We have addressed the concerns raised during the rebuttal and kindly ask if any issues remain unresolved. With the deadline approaching, we would greatly appreciate any further feedback or suggestions at your earliest convenience.
>
> Best regards

---

> ### Author Response · Authors · 2024-12-02
> **Urgent Request for Feedback Before Discussion Closes**
>
> With less than 24 hours remaining before the discussion is closed, we urgently request your feedback. We are confused and upset by the delay, as we believe it is a reviewer’s fundamental responsibility to respond. The discussion phase is a critical part of the ICLR review process, and it is essential for reviewers to engage closely with authors to ensure a fair and objective evaluation of the paper. We have invested significant time addressing your comments, and we are eager to resolve any concerns you may have. Please let us know if there are any further questions. We look forward to your response!

---

> ### Author Response · Authors · 2024-12-03
> **Urgent! Less Than 12 Hours Remaining in Discussion! Your Feedback is STILL Missing!**
>
> Dear Reviewer CTwe,
>
> The discussion phase is nearing its conclusion, with **only 12 hours remaining**, and we are still awaiting your response. This phase is critical for ensuring a thorough and fair evaluation of the paper, and active participation from reviewers is essential to maintain the integrity of the process.
>
> Despite the significant effort we have invested in addressing your comments, your lack of response has hindered progress. **We strongly urge you to provide your feedback immediately to fulfill your responsibilities as a reviewer.**

---

> ### Author Response · Authors · 2024-12-03
> **Less Than 10 Hours Left, Yet No Feedback Received — Urgent Response Needed!**
>
> We are reaching out once again to kindly request your feedback. We are currently uncertain about any challenges that might be hindering your response, and this situation has left us somewhat puzzled. As we hold great respect for the peer review process, we believe it is essential for reviewers who have agreed to participate to engage with authors and provide constructive feedback.
>
> We have dedicated a significant amount of time to thoughtfully addressing each of your comments and suggestions. Throughout this process, we have remained patient, hoping to address any concerns or questions you may have, in a timely manner. Regardless of the final outcome of our submission, we aim for this process to be a positive and enriching experience.
>
> If there are any further questions or clarifications needed, we would be happy to address them. We look forward to your response with anticipation.

---

> ### Comment · Reviewer_CTwe · 2024-12-03
> **Response.**
>
> Thank you for your clarifications. Meanwhile, please kindly notice that the rebuttal addressing my major concern regarding Elo was posted on December 1.
>
> > In the context of decision-making tasks, cyclic dependencies do not arise.
>
> This argument seems less intuitive to me. Decision-making encompasses a broad range of scenarios, and games like rock-paper-scissors are also decision-making problems. While I understand that such games are not the focus of the paper, the fact that non-transitive games are not explicitly handled does not necessarily imply that Elo is unbiased or unaffected in these contexts.
>
> > The quality of a decision sequence can be objectively and unambiguously assessed based on its success rate in completing the final task.
>
> Having a single scalar to evaluate policy performance does not inherently demonstrate that Elo is unbiased. Even in the example of rock-paper-scissors, policy performance can be assessed by its expected rewards, which may vary depending on the opponents faced.
>
> Moreover, I noticed that the revised version introduces a new paragraph mentioning MDPs. If the goal is to solve an MDP, I agree that Elo might not exhibit the issues I raised (though efficiency could become a concern theoretically). However, it remains unclear to me whether MDPs fully capture the application domains considered in the paper—especially in cases where the decision-making environment or dataset collection process may change dynamically due different users.
>
> Therefore, I will adjust my evaluation slightly and decrease my confidence.

---

> ### Author Response · Authors · 2024-12-03
> **Response to Your Feedback**
>
> Thank you for your timely feedback, though I must strongly clarify several points where there seems to be a fundamental misunderstanding regarding the research area of LLM agents.
> 1. **MDP Framework is a Area Consensus**: First and foremost, the decision-making process for large language model agents has been modeled as a Markov Decision Process (MDP) for years. This is not something we "introduced" in this paper, as you suggested, but rather a long-established consensus in the field. From OpenAI’s WebGPT [1] to the latest works [2-18] on LLM agents, MDPs have been the standard framework used to model agent decision-making tasks. Therefore, I must correct the misconception in your comment – we have not deviated from the standard approach or introduced anything new.
> 2. **Real-World Tasks, Not Games**: The focus of current LLM agent research is on solving real-world, multi-step decision-making tasks, not on abstract game-theoretic scenarios like rock-paper-scissors. Tasks such as web browsing [1], software development [13,15], API usage [8-11], embodied simulations [2,3,17], and device control [18] are modeled as MDPs precisely because they reflect real-world decision-making problems where agents are tasked with completing specific goals. The idea that non-transitive games like rock-paper-scissors are in any way comparable to the tasks we are studying is simply misguided. These games involve entirely different dynamics and are not relevant to our work, nor are they within the scope of LLM agent research.
> 3. **Objective Task Performance Evaluation**: In contrast to non-transitive games, LLM agents are evaluated based on their ability to complete specific tasks. The performance of these agents is objectively assessed by their "success rate"—the proportion of tasks completed successfully. This provides a clear, unbiased, and accurate measure of performance. Your suggestion that the single scalar does not demonstrate an unbiased evaluation process is not only incorrect but also entirely detached from the way task-solving performance is evaluated in the LLM agent research community. There is no ambiguity in the success rate metric, which has been the gold standard for evaluating agent performance in this context.
>
> In conclusion, I must to calrify the misleading comments you raised. We are working within the well-established methodologies of the field, and the comparisons you made to rock-paper-scissors or non-transitive games are wholly inappropriate. I expect these points to be reflected in your revised evaluation.
>
> ---
>
> [1] Reiichiro Nakano, et al. WebGPT: Browser-assisted question-answering with human feedback. arXiv 2021.
>
> [2] Mohit Shridhar, et al. ALFWorld: Aligning Text and Embodied Environments for Interactive Learning. ICLR 2021.
>
> [3] Ruoyao Wang, et al. ScienceWorld: Is your Agent Smarter than a 5th Grader? EMNLP 2022.
>
> [4] Shunyu Yao, et al. ReAct: Synergizing Reasoning and Acting in Language Models. ICLR 2023.
>
> [5] Noah Shinn, et al. Reflexion: language agents with verbal reinforcement learning. NeurIPS 2023.
>
> [6] Shunyu Yao, et al. Tree of thoughts: Deliberate problem solving with large language models. NeurIPS 2023.
>
> [7] Shibo Hao, et al. Reasoning with language model is planning with world model. EMNLP 2023.
>
> [8] Minghao Li, et al. API-Bank: A Comprehensive Benchmark for Tool-Augmented LLMs. EMNLP 2023.
>
> [9] Yifan Song, et al. RestGPT: Connecting Large Language Models with Real-World Applications via RESTful APIs. arXiv 2023.
>
> [10] Qin, Yujia, et al. ToolLLM: Facilitating Large Language Models to Master 16000+ Real-world APIs. ICLR 2023.
>
> [11] Chen, Zehui, et al. T-Eval: Evaluating the tool utilization capability of large language models step by step. ACL 2024.
>
> [12] Bilgehan Sel, et al. Algorithm of thoughts: Enhancing exploration of ideas in large language models. ICML 2024.
>
> [13] Weijie Lv, et al. Codeact: Code adaptive compute-efficient tuning framework for code llms. arXiv 2024.
>
> [14] Xiao Liu, et al. AgentBench: Evaluating LLMs as Agents. ICLR 2024
>
> [15] Chen Qian, et al. ChatDev: Communicative Agents for Software Development. ACL 2024.
>
> [16] Yuyan Zhou, et al. MetaGPT: Merging Large Language Models Using Model Exclusive Task Arithmetic. EMNLP 2024.
>
> [17] Peter A. Jansen, et al. DISCOVERYWORLD: A Virtual Environment for Developing and Evaluating Automated Scientific Discovery Agents. arXiv 2024.
>
> [18] Harsh Trivedi, et al. AppWorld: A Controllable World of Apps and People for Benchmarking Interactive Coding Agents. ACL 2024.

---

### Author Response · Authors · 2024-12-04
**General Response**

We would like to express our sincere gratitude to all the reviewers for their valuable time and insightful feedback. In response to their suggestions, we have made several revisions, including additional experiments and clarifications. Below is a summary of the key changes made:
1. We have conducted further experiments with other large language models, including `claude-3-5-haiku` and `llama-3.1-8b`. These results validate the effectiveness of our proposed method across various LLMs (Reviewer v6Tk).
2. Our paper has been revised for improved clarity and precision. This includes adjustments to the positioning of the "Related Works" section, correction of typo errors, and the introduction of formal MDP notations (Reviewer v6Tk).
3. We have analyzed the API call usage of RaDAgent, demonstrating that incorporating Elo score calculations significantly reduces the number of API calls required for decision-making. This highlights the efficiency of our RaDAgent's decision process (Reviewer edTJ).
4. We have clarified the motivation and rationale for selecting the Elo rating system in our agent's decision-making process. Specifically, we emphasize how the inherent characteristics of the Elo algorithm align with the decision-making requirements of the agent (Reviewers edTJ and CTwe).
5. We have highlighted the technical novelty and innovation of adapting the traditional Elo rating system to agent decision-making scenarios (Reviewer CTwe).

Once again, we extend our thanks to all reviewers and area chairs for their invaluable contributions.

---

### Meta-Review · Area_Chair_T7Vg · 2024-12-19

**Metareview:**

This paper introduces a new approach named RaDAgent that addresses the limitation of LLM agents' dependence on external performance measures for decision-making. Instead, the authors propose to use the internal rationality of the agents through an iterative framework combining Experience Exploration and Utility Learning. The framework also incorporates the Elo Rating system to learn and assign utilities, enabling agents to make autonomous decisions based on the learned internal judgment. Overall, the introduction of internal rationality and Elo rating in building LLM agents for decision-making is novel, the experimental results are promising, and the manuscript is overall well-written and easy to follow. The main concerns were regarding the justification of the use of Elo rating system and the lack of theoretical analysis, and the re-organization of certain parts/sections. Overall the paper contains new ideas and results that can benefit the advancement of LLM agents for decision-making. I suggest the authors to incorporate the feedback and new results from the rebuttal in the next version of the paper.

**Additional Comments On Reviewer Discussion:**

There were some concerns regarding the justification of the use of Elo rating system and the lack of associated theoretical analyses, as well as a few clarity/writing issues. The authors acknowledged the comments, and have added new experiments to justify the efficiency in terms of API calls, attributing to the use of Elo rating, and re-organized several sections to improve the clarity/presentation of the paper as suggested. New experiments that justify the effectiveness of the approach across several LLMs were also supplemented in the revision to strengthen the paper. Overall, I found the authors' responses and revisions mostly satisfying.

---

### Decision · Program_Chairs · 2025-01-22

Accept (Poster)